# THE COMPACT SUPPORT NEURAL NETWORK

## ABSTRACT

Neural networks are popular and useful in many fields, but they have the problem of giving high confidence responses for examples that are away from the training data. This makes the neural networks very confident in their prediction while making gross mistakes, thus limiting their reliability for safety critical applications such as autonomous driving, space exploration, etc. In this paper, we present a neuron generalization that has the standard dot-product based neuron and the RBF neuron as two extreme cases of a shape parameter. Using ReLU as the activation function we obtain a novel neuron that compact support, which means its output is zero outside a bounded domain. We show how to avoid difficulties in training a neural network with such neurons, by starting with a trained standard neural network and gradually increasing the shape parameter to the desired value. Through experiments on standard benchmark datasets, we show the promise of the proposed approach, in that it can have good prediction on in-distribution samples, while being able to consistently detect and have low confidence on out of distribution samples.

## 1 INTRODUCTION

Neural networks have been proven to be extremely useful in all sorts of applications, including object detection, speech and handwriting recognition, medical imaging, etc. They have become the state of the art in these applications, and in some cases they even surpass human performance. However, neural networks have been observed to have a major disadvantage: they don't know when they don't know, i.e. don't know when the input is far away from the type of data they have been trained on. Instead of saying "I don't know", they give some output with high confidence (Goodfellow et al., 2015; Nguyen et al., 2015). An explanation of why this is happening for ReLU based networks has been given in Hein et al. (2019). This issue is very important for safety-critical applications such as space exploration, autonomous driving, medical diagnosis, etc. In these cases it is important that the system know when the input data is outside its nominal range, to alert the human (e.g. driver for autonomous driving or radiologist for medical diagnostic) to take charge in such cases.

In this paper we suspect that the root of this problem is actually the neuron design, and propose a different type of neuron to address what we think are its issues. The standard neuron can be written as $f(x) = \sigma(\mathbf{w}^T\mathbf{x} + b)$, which can be regarded as a projection (dot product) $\mathbf{x} \to \mathbf{w}^T\mathbf{x} + b$ onto a direction $\mathbf{w}$, followed by a nonlinearity $\sigma(\cdot)$. In this design, the neuron has a large response for vectors $\mathbf{x} \in \mathbb{R}^p$ that are in a half-space. This can be an advantage when training the NN since it creates high connectivity in the weight space and makes the neurons sensitive to far-away signals. However, it is a disadvantage when using the trained NN, since it can lead to the neurons unpredictably firing with high responses to far-away signals, which can result (with some probability) in high confidence responses of the whole network for examples that are far away from the training data.

To address these problems, we use a type of radial basis function neuron (Broomhead & Lowe, 1988), $f(\mathbf{x}) = g(\|\mathbf{x} - \boldsymbol{\mu}\|^2)$, which we modify to have a high response only for examples that are close to $\boldsymbol{\mu}$, and to have zero response at distance at least $R$ from $\boldsymbol{\mu}$. Therefore the neuron has compact support, and the same applies to a layer formed entirely of such neurons. Using one such compact support layer before the output layer we can guarantee that the space where the NN has a non-zero response is bounded, obtaining a more reliable neural network.

In this formulation, the parameter vector $\boldsymbol{\mu}$ is directly comparable to the neuron inputs $\mathbf{x}$, thus $\boldsymbol{\mu}$ has a simple and direct interpretation as a "template". A layer consisting of such neurons forms can be interpreted as a sparse coordinate system on the manifold containing the inputs of that layer.

Because of the compact support, the loss function of such a compact support NN has many flat areas and it can be difficult to training it directly by backpropagation. However, we will show how to train such a NN, by starting with a trained regular NN and gradually bending the neuron decision boundaries to make them have smaller and smaller support.

The contributions of this paper are the following:

- We introduce a type of neuron formulation that generalizes the standard neuron and the RBF neuron as two extreme cases of a shape parameter. Moreover one can smoothly transition from a regular neuron to a RBF neuron by gradually changing this parameter. We introduce the RBF correspondent to a ReLU neuron and observe that it has compact support, i.e. its output is zero outside a bounded domain.
- The above construction allows us to smoothly bend the decision boundary of a standard ReLU based neuron, obtaining a compact support neuron. We use this idea to train a compact support neural network (CSNN) starting from a pre-trained regular neural network.
- We show through experiments on standard datasets that the proposed CSNN can achieve comparable test errors with regular CNNs, and at the same time it can detect and have low confidence on out-of-distribution data.

## 1.1 RELATED WORK

A common way to address the problem of high confidence predictions for out of distribution (OOD) examples is through ensembles (Lakshminarayanan et al., 2017), where multiple neural networks are trained with different random initializations and their outputs are averaged in some way. The reason why ensemble methods have low confidence on OOD samples is that the high-confidence domain of each NN is random outside the training data, and the common high-confidence domain is therefore shrunk by the averaging process. This reasoning works well when the representation space (the space of the NN before the output layer) is high dimensional, but it fails when this space is low dimensional (see van Amersfoort et al. (2020) for example).

Another popular approach is adversarial training (Madry et al., 2018), where the training set is augmented with adversarial examples generated by maximizing the loss starting from slightly perturbed examples. This method is modified in adversarial confidence enhanced training (ACET) (Hein et al., 2019) where the adversarial samples are added through a hybrid loss function. However, we believe that training with out of distribution samples could be a computationally expensive if not hopeless endeavor, since the instance space is extremely vast when it is high dimensional. Consequently, a finite number of training examples can only cover an insignificant part of it and no matter how many out-of-distribution examples are used, there always will be other parts of the instance space that have not been explored. Other methods include the estimation of the uncertainty using dropout (Gal & Ghahramani, 2016), softmax calibration (Guo et al., 2017), and the detection of out-of-distribution inputs (Hendrycks & Gimpel, 2017). CutMix Yun et al. (2019) is a method to generate training samples with larger variability, which help improve generalization and OOD detection. All these methods are complementary to our approach and could be used together with our classifiers to improve accuracy and OOD detection.

In Ren et al. (2019) are trained two auto-regressive models, one for the foreground in-distribution data and one for the background, and the likelihood ratio is used to decide for each observation whether it is OOD or not. This is a generative model, while our model is discriminative.

A number of works assume that the distance in the representation space (the space of outputs of the last layer before the final classification layer) is meaningful. They will be reviewed next.

Recently, Jiang et al. (2018) proposed a trust score that measures the agreement between a given classifier and a modified version of a $k$-nearest neighbor classifier. While this approach does consider the distance of the test samples to the training set, it only does so to a certain extent since the $k$-NN does not have a concept of "too far", and is also computationally expensive.

A simple method based on the Mahalanobis distance is presented in Lee et al. (2018). It assumes that the observations are normally distributed in the representation space, with a shared covariance matrix for all classes. While we also assume that the distance in the representation space is meaningful, we make a much weaker assumption that the observations for each class are clustered in a number of clusters, not necessarily Gaussian. In our representation, each class is usually covered by more than one compact support neuron, and each neuron could be involved in multiple classes. Furthermore, the method in Lee et al. (2018) simply replaces the last layer of the NN with their Mahalanobis measure

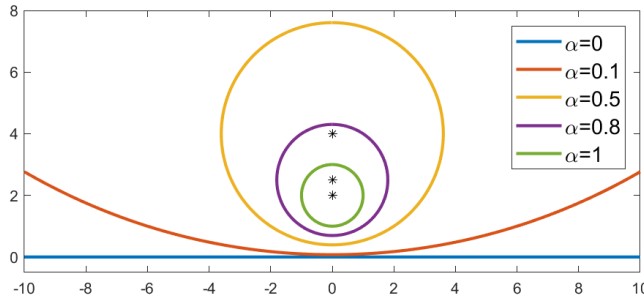

Figure 1: The construction (3) smoothly interpolates between a standard neuron ($\alpha = 0$) and an RBF-type of neuron ($\alpha = 1$). Shown are the neuron decision boundaries for various values of $\alpha$.

and makes no attempt to further train the new model, while we can train our layers together with the whole network.

The Generalized ODIN Hsu et al. (2020) decomposes the output prediction into the ratio of a class-specific function $h_i(x)$ and a common denominator $g(x)$, both defined over instances $x$ from the representation space. Good results are obtained using $h_i$ based on the Euclidean distance or the cosine similarity. Again, this approach assumes that the observations are grouped in a single cluster for each class, which explains it uses very deep models (with 34-100 layers) that are more capable to obtain representations where this assumption is satisfied. Our method does not make the single cluster per class assumption, and can use deep or shallow models.

The Deterministic Uncertainty Quantification (DUQ) (van Amersfoort et al., 2020) method uses an RBF network and a special gradient penalty to decrease the prediction confidence away from the training examples. The authors also propose a centroid updating scheme to handle the difficulties in training an RBF network. In contrast, our paper proposes a generalized neuron model that has the RBF neurons and the standard neurons as two extreme cases, and trains all models starting from a standard NN where the local minima are more well behaved.

## 2 THE COMPACT SUPPORT NEURAL NETWORK

The compact support neural network consists of a number of layers, where the last layer before the output layer contains only compact support neurons, which will be described next. The other layers could be regular neural network or convolutional neural network layers, or compact support layers. The final output layer is a regular linear layer without a bias term, so that it can output a vector of all zeros when appropriate.

### 2.1 THE COMPACT SUPPORT NEURON

We start with the radial basis function (RBF) neuron (Broomhead & Lowe, 1988),

$$f_{\mathbf{w}}(\mathbf{x}) = g(\|\mathbf{x} - \mathbf{w}\|^2). \tag{1}$$

The RBF neuron has $g(u) = \exp(-\beta u)$ as the activation function, but in this paper we will use $g(u) = \max(R^2 - u, 0)$ because it is related to the ReLU.

**A flexible representation.** We can introduce an extra parameter $\alpha = 1$ and rewrite eq. (1) as

$$f_{\mathbf{w}}(\mathbf{x}) = g(\mathbf{x}^T\mathbf{x} + \mathbf{w}^t\mathbf{w} - 2\mathbf{w}^T\mathbf{x}) = g(\alpha(\|\mathbf{x}\|^2 + \|\mathbf{w}\|^2) - 2\mathbf{w}^T\mathbf{x}). \tag{2}$$

Using the parameter $\alpha$, we obtain a representation that smoothly changes between an RBF neuron when $\alpha = 1$ and a standard projection neuron when $\alpha = 0$. However, starting with an RBF neuron with $g(u) = \exp(-\beta u)$, we obtain the projection neuron for $\alpha = 0$ as $f_{\mathbf{w}}(\mathbf{x}) = \exp(2\mathbf{w}^T\mathbf{x})$, which has an exponential activation function.

**The compact support neuron.** We want to obtain a standard ReLU based neuron $f_{\mathbf{w}}(\mathbf{x}) = \sigma(\mathbf{w}^T\mathbf{x})$ with $\sigma(u) = \max(u, 0)$ for $\alpha = 0$. For this purpose we will use $g(u) = \sigma(R^2 - u)$, and modify the above construction to obtain the compact support neuron:

$$f_{\mathbf{w}}(\mathbf{x}) = \sigma(R^2 - \mathbf{x}^T\mathbf{x} - \mathbf{w}^T\mathbf{w} + 2\mathbf{w}^T\mathbf{x}) = \sigma[\alpha(R^2 - \|\mathbf{x}\|^2 - \|\mathbf{w}\|^2 - b) + 2\mathbf{w}^T\mathbf{x} + b], \tag{3}$$

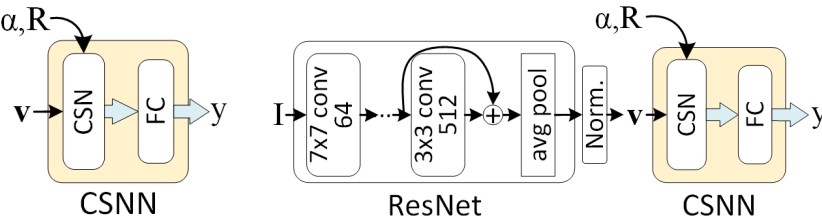

Figure 2: Left: Diagram of the compact support neural network (CSNN), with the CSN layer described in Eq. (6). Right: an example of the CSNN with normalized input from ResNet. Only the full arrows have backpropagation.

where we also introduced a bias term $b$ for the standard neuron. We usually make $b = 0$ for simplicity. The parameter $R$ defines the radius of the support of the neuron when $\alpha = 1$.

One can easily check that the support of $f_{\mathbf{w}}(\mathbf{x})$ from eq. (3) (i.e. the domain where it takes nonzero values) is in a sphere of radius

$$R_\alpha^2 = R^2 + b(1/\alpha - 1) + \|\mathbf{w}\|^2(1/\alpha^2 - 1) \tag{4}$$

centered at $\mathbf{w}_\alpha = \mathbf{w}/\alpha$. Therefore the neuron from eq. (3) has compact support for any $\alpha > 0$ and the larger the value of $\alpha$, the smaller the support of the neuron will be. In Figure 1 is shown the support for several values of $\alpha \in [0, 1]$ of the neuron (3) with $\mathbf{w} = (0, 2)^T, b = 0$ and $R = 1$.

**Convolutional version.** If one desires to make a compact support convolutional neuron, let $\mathbf{w}$ be its $k \times k$ matrix of weights. Then the convolutional version can be obtained by taking into consideration that each $k \times k$ patch of an image $\mathbf{I}$ is a candidate $\mathbf{x}$ in eq. (3). Therefore one can easily check that the convolutional compact support neuron should be:

$$f_{\mathbf{w}}(\mathbf{I}) = \sigma[\alpha(R^2 - b - \mathbf{I}^2 * \mathbf{1} - \|\mathbf{w}\|^2) + 2\mathbf{I} * \mathbf{w} + b] \tag{5}$$

where $\mathbf{1}$ is a $k \times k$ matrix of ones, $\mathbf{I}^2$ is done elementwise and $*$ is the convolution.

## 2.2 THE COMPACT SUPPORT NEURAL NETWORK

If we have a layer containing only compact support neurons (CSN), combining the weights into a matrix $\mathbf{W}^T = (\mathbf{w}_1, ..., \mathbf{w}_K)$ and the biases into a vector $\mathbf{b} = (b_1, ..., b_K)$, we can write the CSN layer as:

$$\mathbf{f_W}(\mathbf{x}) = \sigma(\alpha[R^2 - \mathbf{b} - \mathbf{x}^T\mathbf{x} - \mathrm{Tr}(\mathbf{WW}^T)] + 2\mathbf{Wx} + \mathbf{b}). \tag{6}$$

where $\mathbf{f_W}(\mathbf{x}) = (f_1(\mathbf{x}), ..., f_K(\mathbf{x}))^T$ is the vector of neuron outputs of that layer. This formulation enables the use of standard neural network machinery (e.g. PyTorch) to train a CSN. In practice we will have no bias term (i.e. $\mathbf{b} = 0$), except in low dimensional experiments.

The simplest compact support neural network (CSNN) has two layers: a hidden layer containing compact support neurons (3) or their convolutional counterparts (5), and an output layer which is a standard fully connected layer without bias. It is illustrated in Figure 2, left.

**Normalization.** For best results, all variables of the input data $\mathbf{x}$ should be on the same scale. For better control, it is also preferable that $\|\mathbf{x}\|$ be approximately 1 on the training examples. These goals can be achieved by standardizing the variables to have zero mean and standard deviation $1/\sqrt{d}$ on the training examples (where $d$ is the dimension of $\mathbf{x}$). This way $\|\mathbf{x}\|^2 \sim 1$ when the dimension $d$ is large (under assumptions of normality and independence of the variables of $\mathbf{x}$). Our experiments on three datasets indicate that indeed $\|\mathbf{x}\| \sim 1$ on real data when the inputs $\mathbf{x}$ are normalized as described above, as exemplified by the histograms of $\|\mathbf{x}\|$ from Figure 3.

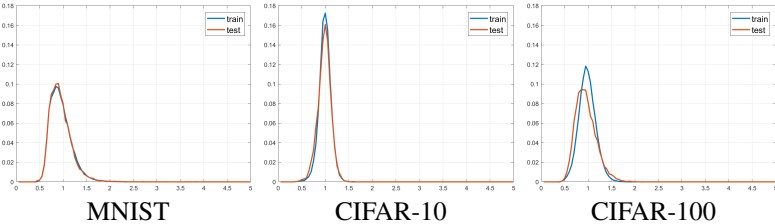

Figure 3: Histogram of the norms $\|\mathbf{v}_i\|$ of the normalized input features $\mathbf{v}_i$ to the CSN layer for the three datasets trained in our experiments.

**Training.** Like the RBF network, training a neural network with such neurons with $\alpha = 1$ is difficult because the loss function has many local optima. To make matters even worse, the compact support neurons have small support when $\alpha$ is close to 1, and consequently the loss function has flat regions between the local minima.

This is why we take another approach to training. Using equations (6) or (5) we can train a CSNN by first training a regular NN ($\alpha = 0$) and then gradually increasing the shape parameter $\alpha$ from 0 towards 1 while continuing to update the NN parameters. Observe that whenever $\alpha > 0$ the NN has compact support, but the support gets smaller as $\alpha$ gets closer to 1. The training procedure is described in detail in Algorithm 1.

---

**Algorithm 1 Compact Support Neural Network (CSNN) Training**

**Input:** Training set $T = \{(\mathbf{x}_i, y_i) \in \mathbb{R}^p \times \mathbb{R}\}_{i=1}^n$,
**Output:** Trained CSNN.

1: Train a regular CNN $\mathbf{f}(\mathbf{x}) = \mathbf{L}\sigma(2\mathbf{W}\mathbf{g}(\mathbf{x}) + \mathbf{b})$ where $\mathbf{W}, \mathbf{L}$ are the last two layer weight matrices and $\mathbf{g}(\mathbf{x})$ is the rest of the CNN.
2: Freeze $\mathbf{g}(\mathbf{x})$, compute $\mathbf{u}_i = \mathbf{g}(\mathbf{x}_i), i = 1, ..., n$, their mean $\boldsymbol{\mu}$ and standard deviation $\boldsymbol{\sigma}$.
3: Obtain normalized versions $\mathbf{v}_i$ of $\mathbf{u}_i$ as $\mathbf{v}_i = (\mathbf{u}_i - \boldsymbol{\mu})/\sqrt{d}\boldsymbol{\sigma}, i = 1, ..., n$.
4: **for** e= 1 to $N^{epochs}$ **do**
5:     Set $\alpha = e/N^{epochs}$
6:     Use the examples $(\mathbf{v}_i, y_i)$ to update $(\mathbf{W}, \mathbf{L}, \mathbf{b})$ based on one epoch of
$$\mathbf{f}(\mathbf{v}) = \mathbf{L}\sigma(\alpha[R^2 - \mathbf{v}^T\mathbf{v} - \text{Tr}(\mathbf{W}\mathbf{W}^T) - b] + 2\mathbf{W}\mathbf{v} + \mathbf{b})$$
7:     (optional) Remove any neurons $\mathbf{w}_j$ of $\mathbf{W}^T = (\mathbf{w}_1, ..., \mathbf{w}_k)$ that are dead, i.e. satisfy:
$$\sigma(\alpha[R^2 - \|\mathbf{v}_i\|^2 - \|\mathbf{w}_j\|^2) - b_j] + 2\mathbf{w}_j^T\mathbf{v}_i + b_j) = 0, i = 1, ..., n$$
8: **end for**

---

In the synthetic experiment in Figure 4 we succeeded to bring the train and test errors close to 0 for $\alpha = 1$ using a carefully crafted schedule for increasing the $\alpha$. However, in the real data applications, the training, test and validation errors might first decrease a little bit but ultimately increase as $\alpha$ approaches 1. For example one could see the test errors vs $\alpha$ for the synthetic dataset in Figure 6 and for the real datasets in Figure 9. For this reason, in practice we stop the training at an $\alpha < 1$ where the training and validation errors still take acceptable values, e.g. a validation error less than the validation error for $\alpha = 0$. However, we noticed that the larger the value of $\alpha$, the tighter the support around the training data and the better the generalization.

It is worth noting that in contrast to the weights of a standard neuron, the weights of the compact support neuron exist in the same space as the neuron inputs and they can be regarded as templates. Thus they have more meaning, and one could easily visualize the type of responses that make them maximal, using standard neuron visualization techniques such as Zeiler & Fergus (2014). Furthermore, one can also obtain samples from the compact support neurons, e.g. for generative or GAN models.

## 3 EXPERIMENTS

In this section we first present an experiment on 2D data to showcase what can be achieved with the proposed compact support neural network, and then experiments on real datasets to show the power of the CSNN to model real data and how it can detect out-of-distribution samples.

### 3.1 2D EXAMPLE

We present a first experiment with the moons 2D dataset, where the data is organized on two intertwining half-circle like shapes, one containing the positives and one the negatives. The data is scaled so that all observations are in the interval $[0, 1]^2$ (shown as a white rectangle in Figure 4. As out of distribution data (OOD) we started with $100 \times 100 = 10000$ samples on a grid spanning $[-0.5, 1.5]^2$ and we removed all samples at distance at most 0.1 from the moons data, obtaining 8763 samples.

We used a two layer CSNN, with the first layer having 128 CSNN neurons, and the second layer being a standard NN layer without bias, as illustrated in Figure 2, left. The second layer is used to integrate

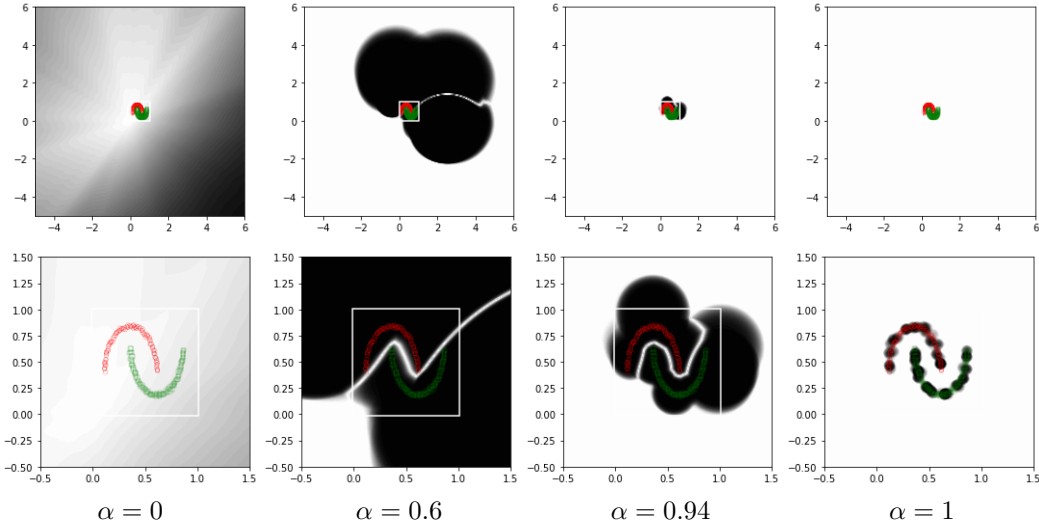

$\alpha = 0$  $\alpha = 0.6$  $\alpha = 0.94$  $\alpha = 1$

Figure 4: The confidence map (0.5 for white and 1 for black) of the trained CSNN on the moons dataset for different values of $\alpha \in [0, 1]$. Top: zoom out on the interval $[-5, 6]^2$. Bottom: zoom in view of the interval $[-0.5, 1.5]^2$.

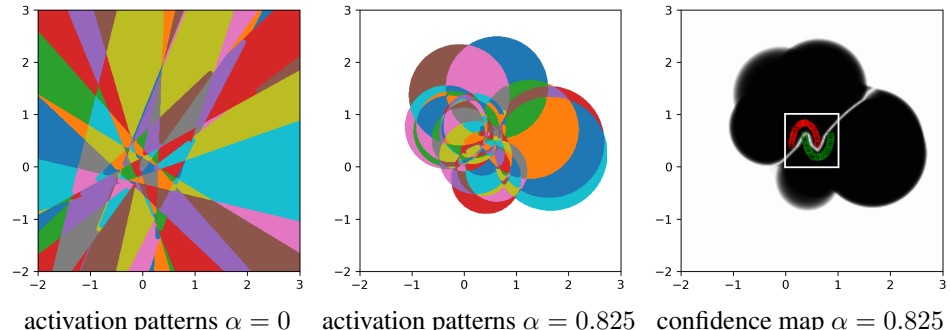

activation patterns $\alpha = 0$  activation patterns $\alpha = 0.825$  confidence map $\alpha = 0.825$

Figure 5: Example of activation pattern domains for $\alpha = 0$ and $\alpha = 0.825$ and the resulting confidence map (0.5 for white and 1 for black) for $\alpha = 0.825$ for a 32 neuron 2-layer CSNN.

the evidence from the CSNN neurons into the class prediction. We used 200 training examples and trained the CSNN using Algorithm 1. We trained 2000 epochs with $R^2$ decreasing linearly from 0.04 to 0.01, and $\alpha$ increasing from 0 to 1 as $\alpha_i = \min(1, \max(0, (i^{0.1} - 1.5)/.6)), i = 1, ..., 2000$. This way $\alpha$ increases slower as it gets closer to 1. Using this special training we avoided the training and test errors blowing up when $\alpha$ gets close to 1. As specified in line 7 of Algorithm 1, the NN nodes that had zero response on all training examples were eliminated. These neurons cannot be trained anymore and only give uncontrolled responses on unseen data. This way from the 128 neurons, only 73 were left at the end of training.

The training/test errors and the AUROC and NZ confidence measures for the OOD data described above vs. $\alpha$ are shown in Figure 6. Observe that the training and test errors for $\alpha = 0$ are quite large, because the standard NN with 128 neurons cannot fit the data well enough, and they decrease as the neuron support decreases and the model is better capable to fit the data.

The confidence map for the obtained classifier is shown in Figure 4. We can see that the confidence is 0.5 (white) almost everywhere except close to the training data, where it is close to 1 (black). This gives us an insight that the method works as expected, shrinking the support of the neurons to a small domain around the training data. We also see that the support is already reasonably small for $\alpha = 0.6$ and it gets tighter and tighter as $\alpha$ gets closer to 1.

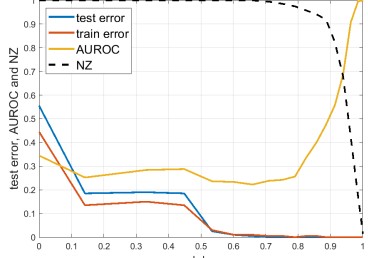

Figure 6: CSNN train and test errors, AUROC and percent nonzero outputs (NZ) vs. $\alpha$ for the moons data.

It is known Croce & Hein (2018); Hein et al. (2019) that the output of a ReLU-based neural network is piecewise linear and the domains of linearity are given by the activation pattern of the neurons. The activation pattern of the neurons consists of the domains where the set of neurons that are active (i.e. their output is positive) does not change. These activation pattern domains are polytopes, as shown in in Figure 5, left, for a two-layer NN with 32 neurons. The activation domains for a CSNN are intersections of circles, as illustrated in Figure 5, middle, with the domain where all neurons are inactive shown in white. The corresponding confidence map is shown in Figure 5, right.

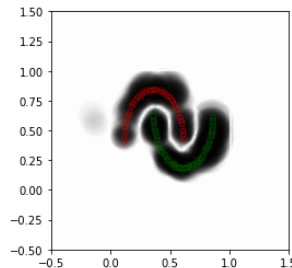

In real data applications we don't need to go all the way to $\alpha = 1$ since even for smaller $\alpha$ the support is still bounded and if the instance space is high dimensional (e.g. 512 to 1024 in the real data experiments below), the volume of the support of the CNN will be very small compared to the instance space, making it unlikely to have high confidence on out-of-distribution data.

**The role of pruning dead neurons.** Due to the random initialization of the neurons, there might exist neurons that have zero response on all the training observations. These neurons are dead in the sense that they are not updated in the back-propagation, since their response is always zero. We have observed that in some cases these dead neurons will produce some small high confidence regions far away from the training examples (see Fig. 7). This problem can be eliminated by removing these neurons during training, which is done by line 7 of Algorithm 1.

Figure 7: Confidence map without pruning, $\alpha = 0.985$.

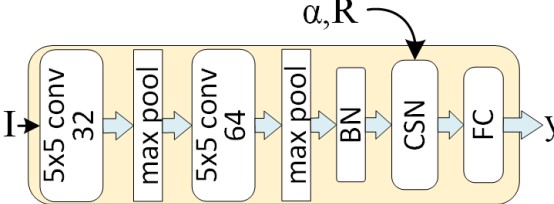

Figure 8: The CSNN-F with LeNet backbone, where all layers are trained by backpropagation.

## 3.2 REAL DATA EXPERIMENTS

We conduct experiments by training on three different datasets: MNIST (LeCun & Cortes, 2010), CIFAR-10 and CIFAR-100 (Krizhevsky et al., 2009). We evaluate the confidence on in-sample and out-of-sample data, by testing them on their respective test sets (in-sample) and on other datasets as shown in Table 1, including the test sets of EMNIST (Cohen et al.), FashionMNIST (Xiao et al., 2017) and SVHN (Netzer et al., 2011), and the validation set of ImageNet (Deng et al., 2009). For MNIST we also tested on a grayscale version of CIFAR-10, obtained by converting the 10,000 test images to gray-scale and resizing them to $28 \times 28$.

**CNN architecture.** For MNIST we use a 4-layer LeNet CNN as backbone, with two $5 \times 5$ convolution layers with 32 and 64 filters respectively, followed by ReLU and $2 \times 2$ max pooling, and two fully connected layers with 256 and 10 neurons. For the other two datasets, we used as backbone a ResNet-18 architecture (He et al., 2016) with 4 residual blocks with 64, 128, 256 and 512 filters respectively. After the backbone CNN has been trained, the FC layers were removed and only the convolutional layers were kept, as illustrated in Figure 2, right and Figure 8.

For the CSNN we will experiment with two architectures, illustrated in Figure 2 and Figure 8. The first is a small one (called CSNN) that takes as input the output of the last convolutional layer of the backbone, normalized as described in Section 2.2. The normalization of the CSNN input can also be achieved using a batch normalization layer without any learnable affine parameters. The second one is a full network (called CSNN-F), illustrated in Figure 8, where the backbone (LeNet or ResNet) is part of the backpropagation and a batch normalization layer (BN) without any learnable parameters has been introduced between the backbone and the CSN layer.

**Training details.** For all datasets we used data augmentation with padding (3 pixels for MNIST, 4 pixels for the rest) and random cropping to train the backbones. For CIFAR-100 we also used random rotation up to 15 degrees. We used no data augmentation when training the CSNN and CSNN-F.

The training/test data was passed though the backbone without the FC layers, and the output was normalized. A CSNN without bias term was trained for 510 epochs with $R = 0.1$, of which 10

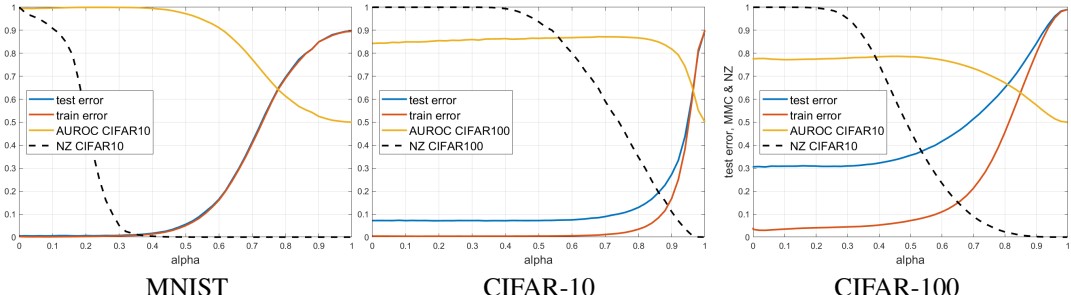

Figure 9: Train and test errors, Area under ROC Curve (AUROC) and percent nonzero outputs (NZ) vs $\alpha$ for CSNN classifiers trained on three real datasets. These results are obtained from one training run.

epochs at $\alpha = 0$. For the CSNN training we used the Adam optimizer with learning rate 0.001 and weight decay 0.0001. We also tried SGD and obtained similar results. The CSNN-F was trained with SGD with a learning rate of 0.001 and weight decay 0.0005. Its layers were initialized with the trained backbone and the trained CSNN. Then $\alpha$ was kept fixed for two epochs and increased by 0.005 every epoch for 4 more epochs.

Training the CSNN from $\alpha = 0$ to $\alpha = 1$ for 510 epochs takes less than an hour on a MSI GS-60 Core I7 laptop with 16Gb RAM and Nvidia GTX 970M GPU. Each epoch of the CSNN-F took less than a minute with the LeNet backbone and about 3 minutes with the ResNet-18 backbone.

**OOD detection.** The out of distribution (OOD) detection is performed similarly to the way it is done in a standard CNN. For any observation, the maximum value of CSNN raw outputs is used as the OOD score for predicting whether the observation is OOD or not. If the observation is in-distribution, its score will usually be large, and if it is OOD, it will be usually close to zero or even zero. The ROC curve based on these scores for the test set of the in-distribution data (as class 0) and one OOD dataset (as class 1) will give us the AUROC. If the two distributions are not separable (have concept overlap), some of the OOD scores will be large, but for the OOD observations that are away from the area of overlap they will be small or even zero.

In Figure 9 are shown the train/test errors vs $\alpha$ for the CSNN on the three datasets. Also shown are the Area under the ROC curve (AUROC) for OOD detection on CIFAR-10 or CIFAR-100 and the percentage of OOD samples with nonzero outputs (NZ). Observe that all curves on the real data are very smooth, even though they are obtained from one run, not averaged. We see that the training and test errors stay flat for a while then they start increasing from a certain $\alpha$ that depends on the dataset. At the same time, the AUROC stays flat and slightly increases, and there is a range of values of $\alpha$ where the test error is low and the AUROC is large. Looking in more detail at the CIFAR-10 dataset (middle plot in Figure 9), we see that for $\alpha = 0.7$ the NZ-CIFAR100 is about 0.6, which means that about 40% of the CIFAR-100 observations have all 0 CSNN outputs, therefore an OOD score of 0. Combined with the scores of the other observations and the fact that at most 10% of the CIFAR-10 test observations have an OOD score of 0 (because the test error is less than 0.1) it results in a slight increase in AUROC.

In practice, $\alpha$ should be chosen as large as possible where an acceptable validation error is still obtained, to have the smallest support possible. For example one could choose the largest $\alpha$ such that the validation error at $\alpha$ is less then or equal to the validation error at $\alpha = 0$. However, for better comparison with the other methods, for each dataset we chose the CSNN classifier corresponding to the largest $\alpha$ where the test error takes a value comparable to the other methods compared, and reported the AUROC values in Table 1. The CSNN-F was obtained by merging the corresponding CSNN head with the ResNet or LeNet backbone and training them together for 6 epochs.

**Methods compared.** We compare our results with the Adversarial Confidence Enhanced Training (ACET) (Hein et al., 2019), Deterministic Uncertainty Quantification (DUQ) (van Amersfoort et al., 2020), a standard CNN, and an ensemble of five or 10 CNNs trained with different random initializations. The ACET results are taken directly from Hein et al. (2019), and the DUQ, CNN and ensemble results were obtained using the DUQ authors' code. For DUQ we trained multiple models with various combinations of the length scales $\sigma \in \{0.05, 0.1, 0.2, 0.3, 0.5, 1.0\}$ and gradient penalty $\lambda \in \{0, 0.05, 0.1, 0.2, 0.3, 0.5, 1.0\}$ and selected the combination with the best test error-AUROC trade-off.

| | CNN | ACET | DUQ | 5 Ens | 10 Ens | CSNN | CSNN-F |
|---|---|---|---|---|---|---|---|
| Train on MNIST | 0.53% (.05) | 0.66 | 0.57 (.04) | 0.50 (.02) | 0.51 (.03) | 0.52 (.01) | 0.50 (0.02) |
| EMNIST | 0.983 (.001) | 0.912 | 0.988 (.001) | 0.985 (.001) | 0.985 (.001) | **0.992** (.001) | 0.990 (.002) |
| FashionMNIST | 0.989 (.001) | **0.998** | **0.998** (.001) | 0.992 (.001) | 0.992 (.001) | **0.998** (.001) | 0.997 (.001) |
| grayCIFAR-10 | 0.995 (.001) | **1.000** | 0.978 (.005) | 0.992 (.003) | 0.992 (.002) | **1.000** (.0001) | **1.000** (.0001) |
| Average | 0.989 (.005) | 0.970 | 0.988 (.009) | 0.990 (.004) | 0.990 (.004) | **0.996** (.003) | **0.996** (.004) |
| Train on CIFAR-10 | 5.99% (.09) | 8.44 | 6.88 (.40) | 4.83 (.16) | 4.59 (.08) | 7.28 (.06) | 6.18 (.09) |
| CIFAR-100 | 0.860 (.001) | 0.852 | 0.827 (.016) | 0.891 (.001) | **0.897** (.001) | 0.865 (.001) | 0.882 (.003) |
| SVHN | 0.899 (.012) | **0.981** | 0.912 (.031) | 0.917 (.006) | 0.924 (.002) | 0.908 (.001) | 0.900 (.013) |
| ImageNet | 0.834 (.002) | 0.859 | 0.816 (.021) | 0.863 (.001) | **0.869** (.001) | 0.848 (.001) | 0.854 (.004) |
| Average | 0.865 (.029) | **0.897** | 0.852 (.050) | 0.890 (.023) | **0.897** (.023) | 0.874 (.026) | 0.879 (.021) |
| Train on CIFAR100 | 26.18% (.28) | 32.24 | 31.14 (.23) | 22.43 (.20) | 21.86 (.11) | 30.89 (0.10) | 24.46 (.12) |
| CIFAR-10 | 0.750 (.002) | 0.720 | 0.722 (.008) | 0.781 (.001) | **0.786** (.001) | 0.783 (.001) | 0.762 (.002) |
| SVHN | 0.781 (.035) | **0.912** | 0.774 (.006) | 0.832 (.013) | 0.834 (.009) | 0.872 (.001) | 0.860 (.006) |
| ImageNet | 0.766 (.002) | 0.752 | 0.742 (.006) | 0.798 (.001) | **0.803** (.001) | 0.755 (.001) | 0.793 (.001) |
| Average | 0.766 (.023) | 0.795 | 0.746 (.024) | 0.804 (.022) | **0.808** (.021) | 0.804 (.050) | 0.805 (.042) |

Table 1: OOD detection comparison in terms of Area under the ROC curve (AUROC) for models trained and tested on several datasets. For each model the test error in % is shown in the "Train on" row. The ACET results are taken from Hein et al. (2019). All other results are averaged over 10 runs and the standard deviation is shown in parentheses.

**Results.** The results are shown in Table 1. All results except the ACET results are averaged over 10 runs and the standard deviation is shown in parentheses. From Table 1 we observe that our methods obtain the best results on MNIST and the 10-ensemble obtains the best results on the other two datasets. The test errors of the CSNN-F approach are smaller than the CSNN, and the AUROCs are comparable. Compared to ACET both CSNN and CSNN-F obtain smaller test errors on all three dataset and better average AUROC on two out of three datasets. Compared to DUQ, the CSNN and CSNN-F obtain comparable test errors and better average AUROC on all three datasets. Compare to the 5-ensemble, the CSNN-F obtains comparable errors on two datasets and comparable or better AUROC on two datasets. Comparing the training time, both our methods are about 4 times faster than training a 5-ensemble, 8 times faster than a 10- ensemble and about 3 times faster than DUQ.

## 4 CONCLUSION

In this paper, we presented a generic neuron formulation that encompasses the standard projection based neuron and the RBF neuron as two extreme cases of a shape parameter $\alpha \in [0, 1]$. By using ReLU as the activation function we obtained a novel type of neuron that has compact support. We showed how to avoid the difficulties in training the compact support NN by training a standard neural network first ($\alpha = 0$) and gradually shrinking the support by increasing $\alpha$. We showed the advantages of the proposed compact support neural network in that it can still have good prediction on data coming from the same distribution, but it can detect out of distribution samples consistently well. This feature is important in safety critical applications such as autonomous driving, space exploration and medical imaging. Our results have been obtained without any adversarial training or ensembling, and adversarial training or ensembling could be used in our framework to obtain further improvements.

In the real data applications we used a compact support layer as the last layer before the output layer. This ensures that the compact support is involved in the most relevant representation space of the CNN. However, because the CNN still has many projection-based layers to obtain this representation space, it means that the corresponding representation in the original image space does not have compact support and high confidence erroneous predictions are still possible. In the future we plan to study architectures with multiple compact support layers that have even smaller support in the image space.

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
