# OpenReview forum: "The Compact Support Neural Network"
_ICLR.cc/2021/Conference — Reject_

### Official Review · AnonReviewer3 · 2020-10-27
**Interesting work that connects RBF neuron and RELU neuron in neural network**

**Rating:** 5
**Confidence:** 3

**Review:**

The main contribution is that the author porposed a formulation which connects the RBF neuron and RELU neuron in neural network to overcome the difficulties in training a DNN model with RBF layer(s).

pros:
- This paper is well written and easy to understand. The illustration summarizes well its effect of OOD predictions.
- The formulation of eqn.3 is interesting and naturally connect RBF and RELU neurons with a parameter $\alpha$.

cons:
- The essential idea of this compact support network is based on the assumption that $L_2$ distance of the representation space is meaningful. Is that correct? How about other metric? How about to pre-calculate the distance on dataset in the representation space, and reject those samples which are away from the class center?
- The performance gain of proposed method comparing to other baseline methods which can also stable the training of RBF network is marginal (espeically in Table.1).
- The backbone and the training datasets of RBF network are quite small, could the proposed method stable the training of large network with large dataset (such as training deeper ResNet on Imagenet)?

---

> ### Author Response · Authors · 2020-11-23
> **Response to Reviewer 3's concerns**
>
> - The essential idea of this compact support network is based on the assumption that distance of the representation space is meaningful. Is that correct? How about other metric? How about to pre-calculate the distance on dataset in the representation space, and reject those samples which are away from the class center?
>
> Yes, the distance is assumed to be meaningful, but the neurons are not centered on the class centers, but are supposed to cover the instance space. In our model one neuron could be involved in multiple classes and multiple neurons are usually involved for one class.
> The assumption that the classes are separable and can be covered by one neuron each is much stronger.
> For example it does not hold for the moons dataset but our model can handle this data without problems.
>
> - The performance gain of proposed method comparing to other baseline methods which can also stable the training of RBF network is marginal (especially in Table.1).
>
> We now added a standard CNN to Table 1. One could see that the DUQ, which is the only other RBF-based method, obtains worse results than the standard CNN on two out of three datasets, while our method obtains the much better results than DUQ or the standard CNN on all three datasets.
>
> - The backbone and the training datasets of RBF network are quite small, could the proposed method stable the training of large network with large dataset (such as training deeper ResNet on Imagenet)?
>
> Yes, because we start from a pretrained deep network. Training just the last 2 layers as a CSNN should take about a day on one GPU for Imagenet. Training the CSNN-F is equivalent to 6 epochs of training the deeper Resnet on ImageNet, starting from the trained deep ResNet and trained CSNN.

---

### Official Review · AnonReviewer1 · 2020-10-28
**Novel algorithm for OOD detection, choice of alpha unclear in practice**

**Rating:** 5
**Confidence:** 4

**Review:**

### Summary

The authors propose a new neural network unit and training algorithm in order to improve OOD detection. Units can smoothly be changed between standard (dot-product) and RBF type through a shape hyperparameter. During training, this hyperparameter is slowly moved in the direction of the RBF shape. Empirical comparisons on three OOD problems are presented, showing that the proposed approach is competitive.


### Score
The authors propose a novel algorithmic idea for OOD detection and show that it is competitive. However, the results show that their approach is highly sensitive to the choice of the shape hyperparameter alpha. In practice, robust ways of choosing and annealing the shape hyperparameter alpha will be needed for the method to be useful. As detailed below, I think this aspect is not addressed in satisfactory detail. Hopefully the weaknesses and comments will be addressed by the authors during rebuttal, currently, I view the paper as marginally below acceptance.

### Pros
+ New methods to reliable detect OOD samples are potentially impactful, since they can be applied across different domains
+ The authors present a novel algorithmic idea to the best of my knowledge
+ CSNNs overall show good performance when compared against alternative approaches

### Cons
- The results show that the authors method is highly sensitive to the choice of the shape hyperparameter alpha. For their empirical comparisons the authors chose alpha as follows: “Based on these insights, for each dataset we chose the classifier corresponding to the largest α where the test error takes a value comparable to the other methods compared, and reported the AUROC and NZ values in Table 1”. This selection strategy can hardly be transferred to practice — it would mean training another OOD method first, in order to select a suitable alpha. For the proposed algorithm to be useful in practice, it is central to devise and demonstrate robust schemes of finding the shape hyperparameter that do not rely on running another method. Else, methods that can be more easily used and show comparable performance, such as deep ensembles, may be preferable despite higher training times
- Similar to the UCQ paper, it would be worthwhile to report performance of larger ensemble methods, e.g., ensemble of 10 networks
- No variability in performance metrics is provided in Table 1, runs should be repeated and reported with different seeds and measures of variability (e.g. the standard error of mean) should be reported

### Additional comments
- My understanding from reading the paper is that neurons with zero outputs are pruned. In figure 9, the amount of neurons with non-zero outputs (NZ curve) seems to quickly go down to 0. Wouldn’t this imply that all neurons in the output layer are pruned, i.e., how can such a network still be used?
- The curves for train and test error of MNIST in Figure 9 seem to be identical, is there an explanation for this?
- Reference section should be revised carefully, e.g.: Goodfellow et al. (2014) was published in ICLR 2015, Hendrycks et al. (2017) misses venue


### Update

The authors have not sufficiently addressed my comments regarding the choice of alpha in practice. While a possible strategy was proposed, it has not been evaluated empirically, see comments below for more details. I am thus maintaining my score.

---

> ### Author Response · Authors · 2020-11-23
> **Response to Reviewer 1's concerns**
>
>
> - The results show that the authors method is highly sensitive to the choice of the shape hyperparameter alpha. For their empirical comparisons the authors chose alpha as follows: “Based on these insights, for each dataset we chose the classifier corresponding to the largest a where the test error takes a value comparable to the other methods compared, and reported the AUROC and NZ values in Table 1”. This selection strategy can hardly be transferred to practice — it would mean training another OOD method first, in order to select a suitable alpha. For the proposed algorithm to be useful in practice, it is central to devise and demonstrate robust schemes of finding the shape hyperparameter that do not rely on running another method. Else, methods that can be more easily used and show comparable performance, such as deep ensembles, may be preferable despite higher training times
>
> We now added a sentence in page 8 about how to choose the alpha parameter in practice and how it relates to our choice for the evaluation.
>
> - Similar to the UCQ paper, it would be worthwhile to report performance of larger ensemble methods, e.g., ensemble of 10 networks
>
> We now report results for both the 5 and 10 ensembles.
>
> - No variability in performance metrics is provided in Table 1, runs should be repeated and reported with different seeds and measures of variability (e.g. the standard error of mean) should be reported
>
> We rerun our experiments and now report results as average of 10 runs, with standard deviation.
>
> Additional comments
> - My understanding from reading the paper is that neurons with zero outputs are pruned. In figure 9, the amount of neurons with non-zero outputs (NZ curve) seems to quickly go down to 0. Wouldn’t this imply that all neurons in the output layer are pruned, i.e., how can such a network still be used?
>
> The NZ measures the nonzero outputs for the OOD data, which we would like to be as few as possible.
> It does not measure the nonzero outputs on the training or test data.
>
> - The curves for train and test error of MNIST in Figure 9 seem to be identical, is there an explanation for this?
>
> The train and test errors are very small but the test error is slightly higher than the training error, as one could see if the image is enlarged. The difference between them is about 0.005, which is very small compared to the range of the plot, from 0 to 1.
>
> - Reference section should be revised carefully, e.g.: Goodfellow et al. (2014) was published in ICLR 2015, Hendrycks et al. (2017) misses venue
>
> We revised the references and fixed a few bugs.

---

> > ### Comment · AnonReviewer1 · 2020-11-24
> > **Choice of alpha still unclear**
> >
> > Thank you for the reply
> >
> > > We now added a sentence about how to choose the alpha parameter in practice and how it relates to our choice for the evaluation.
> >
> > I am assuming that you refer to the following paragraph in the paper:
> >
> > > However, in the real data applications, the training, test and validation errors might first decrease a little bit but ultimately increase as α approaches 1. For example one could see the test errors vs α for the synthetic dataset in Figure 6 and for the real datasets in Figure 9. For this reason, in practice we stop the training at an α < 1 where the training and validation errors still take acceptable values. However, we noticed that the larger the value of α, the tighter the support around the training data and the better the generalization.
> >
> > If so, I still find this problematic: What does "acceptable values" mean in practice? I urge the authors to propose a specific strategy and also use it when reporting their results -- rather than selecting alpha indirectly through test errors of other methods

---

> > > ### Author Response · Authors · 2020-11-25
> > > **Did not submit revision at that time**
> > >
> > > Sorry, we did not upload a revision of the paper at the time that we uploaded the response.
> > > We assumed that the responses will be reviewed after the deadline, so after we will have uploaded the revision.
> > > We have uploaded the revision now and the discussion on the selection of alpha is in page 8.

---

> > > > ### Comment · AnonReviewer1 · 2020-11-25
> > > > **No empirical evaluation of strategy**
> > > >
> > > > Thanks for the clarification.
> > > >
> > > > The section added to the manuscript about the choice of alpha is:
> > > > > In practice, α should be chosen as large as possible where an acceptable validation error is still obtained, to have the smallest support possible. For example one could choose the largest α such that the validation error at α is less then or equal to the validation error at α = 0. However, for better comparison with the other methods, for each dataset we chose the CSNN classifier corresponding to the largest α where the test error takes a value comparable to the other methods compared, and reported the AUROC values in Table 1.
> > > >
> > > > Unfortunately, this is merely a suggestion of how alpha might be chosen in practice. In my opinion, however, it would have been crucial to show how robustly such a strategy performs in empirically. What is more, no validation errors are in figure 9, which makes it impossible to gauge whether such a strategy would be successful.
> > > >
> > > > In light of the authors’ response, I thus do not recommend acceptance of the paper in its current form and am maintaining my score.

---

### Official Review · AnonReviewer2 · 2020-10-28
**This paper proposes compact support neurons for deep networks.**

**Rating:** 6
**Confidence:** 4

**Review:**

In this paper, authors propose compact support neurons to prevent high confidence responses from examples that are away from the training data. The design and training of such a neuron seem novel.

The reviewer has the following comments:

1. In (3), what is R2? The missing definition of R2 adds difficulty to fully interpret subsequent equations.

2. Authors start with the RBF neuron, and then replace it with a compact support neuron. However, the relationship between these two seems not very well-explained.

3. From experiments, the proposed method shows effective for the out-of-distribution (OOD) sample detection task. Authors may like to use cifar-10 and cifar-100 as the example to further explain the out-of-distribution detection. For example, how the conceptual overlap across these two datasets contribute to the results? How does a typical CNN perform for OOD detection?

4. While appreciating proposed OOD detection capability, the reviewer is not fully convinced that a compact support network is in general beneficial to knowledge representation. Of course, such a debate might appear slightly out-of-scope here.

---

> ### Author Response · Authors · 2020-11-25
> **Response to Reviewer2's concerns**
>
> - In (3), what is R2? The missing definition of R2 adds difficulty to fully interpret subsequent equations.
>
> We added a definition of R.
>
> - Authors start with the RBF neuron, and then replace it with a compact support neuron. However, the relationship between these two seems not very well-explained.
>
> We start with the RBF neuron, generalize it to a generic activation function, and then show how the compact support neuron smoothly interpolates between a generalized RBF neuron and a standard projection based neuron.
>
> - From experiments, the proposed method shows effective for the out-of-distribution (OOD) sample detection task. Authors may like to use cifar-10 and cifar-100 as the example to further explain the out-of-distribution detection. For example, how the conceptual overlap across these two datasets contribute to the results?
>
> We added a paragraph explaining how OOD detection is done, and another paragraph discussing the CIFAR-10 example and the relation between the all-zero outputs and OOD detection.
>
> - How does a typical CNN perform for OOD detection?
>
> We now added a column with the results of a typical CNN.
>
> - While appreciating proposed OOD detection capability, the reviewer is not fully convinced that a compact support network is in general beneficial to knowledge representation. Of course, such a debate might appear slightly out-of-scope here.
>
> We believe it is quite important for knowledge representation to know that there is a way to interpolate between a RBF neuron and a standard neuron. We also find it interesting to see what is the correspondent to a ReLU for an RBF neuron through this interpolation.

---

### Official Review · AnonReviewer4 · 2020-10-28
**The**

**Rating:** 6
**Confidence:** 3

**Review:**

The paper presents an approach that supports better performance when out of distribution cases occur. It does so by letting neurons be of only compact support and thus if the input is out of distribution (OOD) it is expected to be outside that support and therefore the output will be zero. This is used to detect OOD examples. A parameter alpha is used in the algorithm that determines the size of the support. When it is small then the network acts very similarly to a regular network and when it increases the support is limited. Therefore, to make training stable, they start with a small alpha and then increase it throughout the training.
Then in inference, a value for alpha should be selected that balances the classification accuracy and the OOD detection. For small alpha the classification accuracy is relatively good, yet, the OOD detection is virtually zero. When alpha is very large the OOD detection rate is very high but the classification accuracy deteriorates significantly.

The strategy proposed in this work is quite interesting and might be useful. Yet, I have some concerns:

To make the comparison fair for all the methods, the same test error should be used. As can be seen from the graphs, the OOD decreases significantly when the Test error decreases.

Also, a discussion on the selection of alpha is missing. How it should be selected in a real application?

Another issue is related to some recent works

How this work compares to
-Kimin Lee, Kibok Lee, Honglak Lee, and Jinwoo Shin. A simple unified framework for detecting out-of-distribution samples and adversarial attacks. In International Conference on Neural Information Processing Systems, 2018


-https://proceedings.neurips.cc/paper/2019/hash/1e79596878b2320cac26dd792a6c51c9-Abstract.html

- https://openaccess.thecvf.com/content_CVPR_2020/html/Hsu_Generalized_ODIN_Detecting_Out-of-Distribution_Image_Without_Learning_From_Out-of-Distribution_Data_CVPR_2020_paper.html


-https://openaccess.thecvf.com/content_ICCV_2019/html/Yun_CutMix_Regularization_Strategy_to_Train_Strong_Classifiers_With_Localizable_Features_ICCV_2019_paper.html


Review Update:
I am raising my score but I think the following changes are still required in the paper:
1. Add a section about the choice of alpha (current addition is lacking) discussing the robustness of alpha selection and how easy it is to select it on real problems.
2. Compare to the methods mentioned.
3. Display the proposed approach on deep networks. Does the proposed approach apply there? Given that current SOTA uses very deep networks (with skips) it is important to be able to use the proposed method with such networks.

---

> ### Author Response · Authors · 2020-11-23
> **Response to Reviewer 4's concerns**
>
> - To make the comparison fair for all the methods, the same test error should be used. As can be seen from the graphs, the OOD decreases significantly when the Test error decreases.
>
> Different methods obtain different errors on the same dataset and they have been compared as such in other papers.
> Some methods simply cannot obtain the same test errors as the best one or cannot be tuned to obtain different AUROC/accuracy trade-offs.
> So we chose to obtain a test error that is as close to the best as we can, and better than the worst error of the other methods compared.
>
> - Also, a discussion on the selection of alpha is missing. How it should be selected in a real application?
>
> We added a sentence in page 8 about how to choose alpha in practice.
>
> - How this work compares to
> -Kimin Lee, Kibok Lee, Honglak Lee, and Jinwoo Shin. A simple unified framework for detecting out-of-distribution samples and adversarial attacks. In International Conference on Neural Information Processing Systems, 2018
>
> Lee et al 2018 uses the Mahalanobis distance with a common covariance matrix to obtain the class prediction. This assumes that the observations are grouped in a single cluster for each class in the representation space, which is stronger than our assumption that they are just clustered together in a number of clusters.
>
> -https://proceedings.neurips.cc/paper/2019/hash/1e79596878b2320cac26dd792a6c51c9-Abstract.html
>
> Ren et al 2019 trains two autoregressive models, one for the foreground and one for the background and uses the log-likelihood ratio for OOD detection. Their model is generative while our model is discriminative.
>
> -https://openaccess.thecvf.com/content_CVPR_2020/html/Hsu_Generalized_ODIN_Detecting_Out-of-Distribution_Image_Without_Learning_From_Out-of-Distribution_Data_CVPR_2020_paper.html
>
> Hsu et al 2020 decomposes the output prediction into the ratio of a class specific function and a common denominator, with inputs from the representation space. They obtain good results when the numerator is based on the Euclidean distance or cosine similarity. This again assumes that observations are grouped into a single cluster for each class, which is a stronger assumption than ours.
>
> -https://openaccess.thecvf.com/content_ICCV_2019/html/Yun_CutMix_Regularization_Strategy_to_Train_Strong_Classifiers_With_Localizable_Features_ICCV_2019_paper.html
>
> CutMix (Yun et al 2019) is a method for generating better training examples, and thus is complementary to our representation.
>
> We added a review of these papers to the related work. We did not have enough time to perform experiments with any of these methods on our datasets.

---

### Decision · Program_Chairs · 2021-01-07
**Final Decision**

**Decision:**

Reject

**Comment:**

The paper presents an approach that supports better performance when out of distribution cases occur, by letting neurons be of only compact support and thus if the input is out of distribution (OOD).

Pros:
- The proposed strategy is interesting and may be useful.

Cons:
- The choice of the parameter alpha, whose value is crucial to the success in experiments, is left murky. The approach suggested by the authors was not validated experimentally.
- There is insufficient comparison to recent works.